# A Qualitative Study Exploring Motivators and Barriers to COVID-19 Vaccine Uptake among Adults in South Africa and Zimbabwe

**DOI:** 10.3390/vaccines11040729

**Published:** 2023-03-25

**Authors:** Nellie Myburgh, Mamakiri Mulaudzi, Gugulethu Tshabalala, Norest Beta, Kimberley Gutu, Stefanie Vermaak, Charles Lau, Catherine Hill, Lawrence Stanberry, Wilmot James, Shabir Madhi, Tariro Makadzange, Janan Janine Dietrich

**Affiliations:** 1Vaccines and Infectious Disease Analytics (VIDA) Research Unit, Faculty of Health Sciences, University of the Witwatersrand, Johannesburg 2000, South Africa; 2African Social Sciences Unit of Research and Evaluation (ASSURE), Wits Health Consortium, Faculty of Health Sciences, University of the Witwatersrand, Johannesburg 2000, South Africa; 3Perinatal HIV Research Unit (PHRU), School of Clinical Medicine, Faculty of Health Sciences, University of the Witwatersrand, Johannesburg 2000, South Africa; 4Charles River Medical Group, Harare, Zimbabwe; 5GeoPoll, 3000 Lawrence Street, Suite 125, Denver, CO 80205, USA; 6Department of Pediatrics, Vagelos College of Physicians and Surgeons, Columbia University, New York, NY 10032, USA; 7Institute for Social and Economic Research and Policy, Columbia University, New York, NY 10027, USA; 8Health Systems Research Unit, South African Medical Research Council, Bellville 7538, South Africa

**Keywords:** COVID-19, SARS-CoV-2, vaccines, hesitancy, Africa

## Abstract

While vaccines are a well-established method of controlling the spread of infectious diseases, vaccine hesitancy jeopardizes curbing the spread of COVID-19. Through the Vaccine Information Network (VIN), this study explored barriers and motivators to COVID-19 vaccine uptake. We conducted 18 focus group discussions with male and female community members, stratified by country, age group, and—for Zimbabwe only—by HIV status. Participants’ median age across both countries was 40 years (interquartile range of 22–40), and most (65.9%) were female. We conceptualized the key themes within the World Health Organization’s Strategic Advisory Group of Experts on Immunization (SAGE) 3C (convenience, confidence, complacency) vaccine hesitancy model. Barriers to vaccine uptake—lack of convenience, low confidence, and high complacency—included inaccessibility of vaccines and vaccination sites, vaccine safety and development concerns, and disbelief in COVID-19’s existence. Motivators to vaccine uptake—convenience, confidence, and low complacency—included accessibility of vaccination sites, user-friendly registration processes, trust in governments and vaccines, fear of dying from COVID-19, and knowing someone who had died from or become infected with COVID-19. Overall, vaccine hesitancy in South Africa and Zimbabwe was influenced by inconvenience, a lack of confidence, and high complacency around COVID-19 vaccines.

## 1. Introduction

The global coronavirus (COVID-19) pandemic has had far-reaching negative effects on the daily lives of most people across the world, including southern Africa [1]. Globally, more than 550 million confirmed cases of COVID-19 have been reported [2]. As of 21 September 2022, South Africa had recorded at least 4,016,081 confirmed cases and 102,169 COVID-19-related deaths [3]. Zimbabwe’s numbers were lower, with 257,156 confirmed cases and 5598 COVID-19-related deaths [4]. Recognizing the negative health impacts of COVID-19, South Africa and Zimbabwe swiftly delivered vaccines to their citizens [5]. While vaccines are a well-established method of controlling the spread of infectious diseases, vaccine hesitancy posed a direct threat to curbing the spread of COVID-19. The COVID-19 vaccine rollout began in February 2021 in South Africa and Zimbabwe. Both countries used a staggered approach, first targeting healthcare workers and high-risk individuals, before making the vaccine available to the rest of the population [5]. Since then, over 35 million total doses have been administered in South Africa, and over 11 million doses have been administered in Zimbabwe [5]. Although these numbers are encouraging, vaccination rates have been less than satisfactory, with South Africa and Zimbabwe failing to reach the World Health Organization (WHO)’s goal to vaccinate 70% of the population by the end of 2022 [6]. 

Vaccines are developed to prevent the spread of an infectious disease and, in the process, achieve herd immunity in a population. COVID-19 vaccines recognized by the WHO include Pfizer–BioNTech, Oxford–AstraZeneca, Sinopharm BIBP, Moderna, Janssen, CoronaVac, Covaxin, Novavax, Convidecia, and Sanofi–GSK [7]. At the time of writing, Seven others were under assessment by the WHO: Sputnik V, Sinopharm WIBP, Abdala, Zifivax, Corbevax, COVIran Barekat, and SCB-2019 [8]. 

Vaccine hesitancy, defined as “a delay in acceptance or refusal of vaccination despite the availability of vaccination services” [9], warrants global attention. The WHO’s Strategic Advisory Group of Experts on Immunization (SAGE) working group identified three concepts influencing vaccine hesitancy and uptake, known as the 3Cs: convenience (accessibility), confidence (trust, safety, and effectiveness), and complacency (value of and need for vaccination) [10] (See Figure 1). In this study, we used the 3C model as opposed to the 5C model, which includes confidence, complacency, constraints, calculations, and collective responsibility [11]. We did not use the more recent 7C model, which includes confidence, complacency, constraints, calculation, collective responsibility, compliance, and conspiracy [12].

Several factors influence an individual’s decision to get vaccinated, including vaccine-related misconceptions, mistrust in vaccine benefits, disease-related perceptions, and individual beliefs or superstitions [13,14]. Key themes raised repeatedly in qualitative research into COVID-19 vaccine hesitancy are the unprecedented speed of the disease’s co-evolutionary development and its effect on vaccine safety, potential side-effects and adverse events, the time–cost tradeoffs of vaccination, and questioning the necessity to receive the vaccine or boosters [13]. Additional factors in the African context contributing to lower-than-anticipated COVID-19 vaccine uptake include a young, mobile population, a large, informal job sector, and periodic social unrest [15]. For example, in South Africa, young people were more likely to be vaccine-hesitant, probably owing to rumors concerning the dangers of COVID-19 vaccines and other conspiracy theories circulating on social media and directed at younger audiences [16]. 

Public trust in vaccines demands in-depth community engagement and understanding of contextual factors affecting vaccine uptake [15], allowing for targeted and effective health communication strategies. The Vaccine Information Network (VIN) study addressed the motivators for and barriers to COVID-19 vaccination, using the information it gathered to develop communications to strengthen vaccine uptake in South Africa and Zimbabwe [9]. This paper explores the motivators and barriers to COVID-19 vaccination by applying the SAGE 3Cs framework in a South African and Zimbabwean context.

## 2. Materials and Methods

### 2.1. Study Design

We utilized an exploratory qualitative research design by conducting 18 focus group discussions (FGDs) in Zimbabwe and South Africa from 23 February to 3 March 2022 and from 25 February to 20 May 2022, respectively. Our primary aim was to understand the motivators and barriers to COVID-19 vaccine uptake among adults. Data were collected from both men and women ranging in age from 18 to 60+ years.

### 2.2. Study Setting

At the time of the qualitative data collection, South Africa was at adjusted alert level 1 [13] of the national COVID-19 lockdown restrictions, which required mandatory wearing of face masks in public spaces (including public transport), 50% maximum capacity in venues for indoor and outdoor gatherings, and physical distancing of 1 m in all settings, except schools [13]. We collected data in two provinces—Gauteng and the Western Cape—both of which were experiencing high rates of vaccine hesitancy [17] at the time of data collection. 

While collecting data in Zimbabwe, conditions were slightly more relaxed than in South Africa, although masks were still required, and temperature checks, hand sanitizing, and proof of vaccination remained mandatory for entering certain premises. Businesses reverting to full or partial in-person operations had to adhere strictly to the wearing of masks, temperature checks, sanitizing of hands, and social distancing. Contact classes commenced at schools and other educational institutions, subject to compliance with recommended sanitary conditions [15].

### 2.3. Sampling

We used purposive [18] and chain referral [19] sampling to approach and recruit eligible participants (18 years and older). Unvaccinated community members were approached within their communities, in and around malls, and at taxi ranks [20]. Vaccinated community members, on the other hand, were recruited mainly at vaccination sites: Zimbabwe—in clinics in Harare, including their two affiliated outreach sites; South Africa—from four vaccination sites in Gauteng and five in the Western Cape. 

### 2.4. Procedures 

A total of 18 FGDs were conducted, with a minimum of seven participants each in 15 of the 18 FGDs while three FGDs had eight participants each. FGDs lasted approximately 1 h in Zimbabwe and 2 h in South Africa. In South Africa, we conducted 12 FGDs with vaccinated and unvaccinated participants in Gauteng and the Western Cape (six in each province). In Zimbabwe, we conducted six FGDs with vaccinated and unvaccinated participants; one of the six FGDs was with unvaccinated individuals living with HIV. This stratification was based on the local context and what was deemed appropriate according to the local study team’s feedback. Therefore, in Zimbabwe, a separate group was formed for individuals living with HIV; however, in South Africa, FGDs were not stratified by HIV status, owing to the stigma related to HIV. Focus group discussions allowed for collection of contextual data to gain an in-depth understanding of the motivators and barriers to COVID-19 vaccine uptake [21]. 

### 2.5. Measures

All participants completed a sociodemographic questionnaire assessing age (Table 1), gender, level of education, language spoken at home, and employment status (Table 2). Trained interviewers used a semi-structured guide to facilitate discussions on barriers to and motivators for COVID-19 vaccination, and community perceptions around COVID-19 vaccines. 

### 2.6. Data Analysis

The FGD audio recordings were transcribed verbatim and translated into English for data analysis. Two analysts conducted the initial coding on the basis of the interview guide questions. After developing a coding framework, the two analysts immersed themselves in the data by reading through four of the same transcripts independently [22]. Thereafter, they coded all transcripts using the coding framework and added emergent codes until no new codes emerged. Both data analysts then met to discuss differences and similarities in the codes identified until agreement was reached. Codes were then sorted, collated, and categorized into themes and subthemes. Codes were captured using Nvivo 11 data analysis software, and a comprehensive codebook was developed. The initial codebook was shared with the Zimbabwe research team for review and approval. The final codebook was reviewed and approved by J.J.D., T.M., and N.M. 

The SAGE 3C model of vaccine hesitancy was applied after coding the data. Data analyses were stratified by country, age groups (South Africa: 18–35, 36–49, and 50 years+; Zimbabwe: 18–44 and 45 years+), vaccination status, and—for Zimbabwe only—by HIV status. Data analyses in South Africa were further stratified by province (Gauteng and the Western Cape). 

### 2.7. Ethical Considerations

Study procedures were approved by the University of the Witwatersrand and the University of Cape Town for South Africa, and by the Medical Research Council of Zimbabwe. All participants provided voluntary written informed consent prior to the study and were reimbursed for transport and their time, in line with each country’s specifications (SA: R150 (~9 USD) and Zimbabwe 10 USD). 

## 3. Results

### 3.1. Participants’ Sociodemographics

The median age for participants across both countries was 40 years (interquartile range of 22–40). Most of the 129 participants (65.9%, n = 85) were female, and 87.6% (n = 113) were Black African. Sixteen (12.4%) identified as Colored (In Brown, 2000, p 198: “any person of ‘mixed blood’ including children as well as descendants from Black–White, Black–Asian, White–Asian, and Black–Colored unions” is an officially acceptable racial classification referred to as “Colored”). Multiracial ethnic communities in South Africa may have had ancestors from more than one population inhabiting the region—including Malaysian slaves, slaves from other ethnic groups, Europeans, Africans, or a combination —but were all from South Africa [23]. In the South African context, the classifications of Black African and Colored are not racial but rather social constructs of South African historical apartheid origin [24]. 

Almost 88% of the study participants (87.5%, n = 113) were in the 20–59 year age group, representing the majority of the study population for both South Africa (n = 75) and Zimbabwe (n = 38). More than half of the study participants (59.7%, n = 77) were unemployed, most of whom were from South Africa (77.8%, n = 63 vs. 29.2%, n = 14). The level of education varied by country; in South Africa, over 70% had completed secondary school (grade 12) and, in Zimbabwe, about 66% had completed their O levels (O levels are the Zimbabwean equivalent of grade 12 in South Africa). In Zimbabwe, 16.7% (n = 8) of participants reported living with HIV (See Table 2). 

Barriers to and motivators for COVID-19 vaccination, as illustrated in Figure 2, are discussed using the SAGE 3C model of vaccine hesitancy as a framework, with the boxes under each of the 3Cs (confidence, convenience, and complacency) representing the study themes [10].

**Table 2 vaccines-11-00729-t002:** Participant demographics. The bold indicates the Variable headings.

Variables	South Africa (n = 81)	Zimbabwe (n = 48)	Total (n = 129)
**Gender**			
Female	56 (69.1%)	29 (60.4%)	85 (65.9)
Male	25 (30.9%)	19 (39.6%)	44 (34.1)
Median age (IQR):	40 (23)	39.5 (19)	40 (22)
**Race**			
Black	65 (80.2)	48 (100)	113 (87.6)
Colored	16 (19.8)		16 (12.4)
**HIV status** Negative Positive	* Not assessed	40 (83.3) 8 (16.7)	N/A
**Education**			
Primary	9 (11.1)	3 (6.2)	12 (9.3)
Secondary	61 (75.3)	9 (18.8)	70 (54.3)
O level	–	32 (66.7)	32 (24.8)
A level	–	3 (6.2)	3 (2.3)
Post matric diploma	9 (11.1)	1 (2.1)	10 (7.8)
Bachelor’s degree	2 (2.5)	–	2 (1.6)
**Employment**			
Unemployed	63 (77.8)	14 (29.2)	77 (59.7)
Self-employed	2 (2.5)	22 (45.8)	24 (18.6)
Employed part time	3 (3.7)	2 (4.2)	5 (3.9)
Volunteer work	7 (8.6)	1 (2.1)	8 (6.2)
Employed full time	6 (7.4)	8 (16.6)	14 (10.9)
Pensioner	–	1 (2.1)	1 (0.7)
**Vaccination status**			
Unvaccinated	37 (45.7)	24 (50)	61 (49.3)
Vaccinated	44 (54.3)	24 (50)	68 (52.7)

* South Africa did not have an FGD for participants living with HIV. For the Variables column, the variable headings are in bold.

### 3.2. Barriers to COVID-19 Vaccination

Participants identified several barriers to COVID-19 vaccine uptake, including inaccessibility of vaccines and vaccination sites, and vaccine safety concerns. The uptake of COVID-19 vaccinations was also influenced by the fear of side-effects, conspiracy theories, and concerns about the pace of development of COVID-19 vaccines. 

#### 3.2.1. Lack of Convenience as Barrier 

Participants identified restricted access to vaccines, including the scarcity of vaccinations per site, waiting in long queues, and government corruption as barriers to vaccine uptake. 

Long queues at vaccination sites. A frequent challenge in both South Africa and Zimbabwe, some vaccinated participants across all age groups in Gauteng and unvaccinated participants aged 18–35 years in the Western Cape reported standing in queues and waiting long periods before receiving their vaccines: 

“At clinics it [will] be full, that made people not to go. It would be too full.” 
*(V_50+ GP)*


“If there’s a line, so most of the people isn’t going because no one wants to stand in the line.” 
*(UV_18–35_WC)*


Additionally, overcrowding, long queues, and prolonged waiting times at the vaccination stations impelled Zimbabweans across all groups to abandon their attempts at being vaccinated: 

“There were a lot of people at the vaccination site to an extent that someone wakes up around 4 am but could not get the number to be vaccinated. It’s true there were now many people who knew they now needed to be vaccinated. Without blaming the nurses, some people lost interest and eventually stayed at home.” 
*(V_45+_ZIM)*


To mitigate the challenge of long queues at vaccination sites impinging on their daily commitments, some participants woke up sufficiently early to secure a place at the front of the queue. One participant reported: 

“I just think what’s preventing people… in fact, from the time you enter through the gate here going to the point where they vaccinate you, maybe I’m expecting to spend 30 min. After spending the 1 h here, I will also tell people back home that I almost died of hunger there at the hospital whilst they linger and vaccinate us at their own pace, so the ones I tell will hesitate to come.” 
*(V_45+_ZIM)*


Capped number of vaccines and vaccinations per sites. Structural challenges experienced at facilities included—in the case of Zimbabwe—limits to the number of people who could be vaccinated per day. Some people would wait the entire day without receiving their vaccination and would be advised to return the following day. Unsurprisingly, some people became discouraged and did not return, as illustrated in the following statement:

“… Imagine waking up early in the morning, feeling cold for 2 days and you don’t get the vaccine because their number is enough for that particular day. When it happened to some people, they lost interest and stayed at home forever.” 
*(V_45+_ZIM)*


In South Africa, waiting times for vaccinations depended on when a batch was opened, which only happened once sufficient people had arrived to allow for the entire batch to be used. One participant who had the unpleasant experience of having to wait for a vaccine batch to be opened, although prepared to wait, felt that people who are hesitant to vaccinate may be reluctant to do so: 

“… But when I returned for the second time, the challenge was [that] there were only two of us, so we had to wait for an additional four people to come so that they could open a batch of vaccines and inject us. We came in early, and we had to wait close to an hour, waiting for people trickling in. That one was also the challenge.” 
*(V_50+_GP)*


Buying of vaccination certificates. In Zimbabwe, participants across both age categories reported that community members bought vaccine certificates or paid bribes to skip the long queues at the vaccination sites. Most indicated that paying health workers in exchange for a vaccination certificate without getting vaccinated was common practice: 

“Some people said, ‘we don’t get the vaccine; we buy vaccination cards’. So, I feel like the whole system was bad.”
*(V_45+_ZIM)*


“Some would come and stand at the gate with money on a paper and hand it to someone … That discouraged me because I will have come in time. The person who would’ve come will just hand over money and do not get vaccinated and leave you in the queue pretending as if they have been vaccinated.” 
*(V_18-44_ZIM)*


#### 3.2.2. Low Confidence as a Barrier

Mistrust in the safety of vaccines, the pharmaceutical companies providing or encouraging vaccinations, and community rumors about why vaccines were developed were indicators of low confidence in the vaccine. Low confidence and uncertainty in this study may have given rise to vaccine hesitancy among some participants.

Vaccine safety concerns. Most participants in both countries reported that COVID-19 vaccine safety concerns were a major barrier, irrespective of their vaccination status and age category. Safety concerns included fears around expiry dates, the biochemical composition of vaccines, their efficacy, and side-effects, and given that vaccines were still undergoing clinical trials. Some participants between the ages of 18 and 49 years—mostly from the unvaccinated FGDs in Gauteng—expressed concerns with the vaccine expiry date:

“We also discovered on the news that when the first batch arrived—I think it’s Johnson & Johnson or Pfizer—they told us that the vaccines expired, so I think a lot of people got skeptical from that point, because how can they order things [vaccines] that are expired?” 
*(UV_18–35_GP)*


Participants expressed serious misgivings about the COVID-19 vaccines, possibly stemming from their concerns with the speed at which they were developed and made available in both countries. 

The pace of development of the vaccine. Vaccinated and unvaccinated participants in both countries—although mostly the 35–49-year-olds in South Africa—questioned the unprecedented rapid pace of the development of COVID-19 vaccines. One participant stated:

“But this thing, COVID, only just arrived but it already has a vaccine, which doesn’t make sense.” 
*(UV_35–49_SA)*


Some participants from Gauteng in the 18–35 and 50+ age groups expressed concern about the vaccines’ effectiveness considering they were still undergoing trials. A few participants also questioned why this vaccine had been given priority over equally devastating epidemic or pandemic diseases and conditions such as HIV/AIDS, Ebola, cancer, and others:

“… Cancer and HIV have always been here but now there is COVID-19 and they managed to get the vaccine so quick. When did they do the research? So quick there is vaccine, and they collect it from the airport. When did they manufacture the vaccine? There is this disease, but already they manufactured this. When did they mix the medications and see how they function? They cannot get the cure for HIV/AIDS, cancer, high blood, and sugar [diabetes].” 
*(V_35–49_11_GP)*


Conspiracy theories. Concerns about the remarkably rapid development of vaccines may have contributed to the high prevalence of conspiracy theories to which (mostly unvaccinated) participants from both countries subscribed. Participants claimed that the vaccine had been developed by the government, funders, pharmaceutical companies, and wealthy, influential people to reduce the population:

“Even now there are those who say the government sees that there are a lot of us, and this is the way to kill us.” 
*(V_50+_GP)*


In South Africa, rumors emanated mainly from unvaccinated participants who claimed that their bodies were being controlled, and that people would be turned into zombies, as magnets were implanted in them during the vaccination process:

“… But people are saying we are going to change in 2 or 3 years and become zombies, all those who are vaccinated. I am so scared to be vaccinated and I am not ready yet.” 
*(UV_35–49_WC)*


In Zimbabwe, the rumors heard by 18–44 year old vaccinated and unvaccinated participants were that people who were vaccinated would “wake up” with deformities, or that the COVID-19 vaccine had been manufactured to kill people to reduce the population. One participant stated:

“So, this is why we couldn’t get vaccinated because of what we heard in the community—people saying ‘Ahh, this vaccine was brought to clear [reduce] the population, it’s clearing and it’s killing’.” 
*(UV_18–44_ZIM)*


Conspiracy theories about the COVID-19 vaccination are not surprising given the participants’ reservations about their governments and their perceptions that politicians were imposing COVID-19 vaccines on people.

Fear of side-effects and adverse events among people living with chronic conditions. Some participants living with HIV from Zimbabwe and across all FGDs in South Africa and Zimbabwe were afraid of the unknown side effects and potential adverse events they may experience subsequent to vaccinating. These fears were a consequence of knowledge gaps about COVID-19 vaccines, antiretroviral therapy (ART), and community rumors. One participant stated:

“… we [participants in the FGD] take ART, so with the news that you may die, the vaccine will destroy all the drugs you are taking, so the vaccine is too powerful, so we said, ‘let’s wait’… plus we are on ART, and in some other times we have challenges with the ART… so when I have problems after getting vaccinated, people will start blaming the vaccine, but I may have been terminally ill for long.” 
*(UV_R_18–44_ZIM)*


Participants’ uncertainties about the safety of the COVID-19 vaccine and the speed with which it became available may also have contributed to their fear of experiencing side-effects.

#### 3.2.3. High Complacency as a Barrier

High levels of complacency were reported by unvaccinated participants in both countries and across all age categories. High complacency was identified in participants who were unvaccinated and had perceptions that COVID-19 was a fabricated disease and did not actually exist. High complacency meant that participants underplayed the gravity and risk of death by COVID-19. 

The prevalence of conspiracy theories and misinformation may have given rise to misconceptions in communities that COVID-19 did not exist. This type of narrative was present only in discussions with unvaccinated participants across all age groups in South Africa. Some unvaccinated participants in Zimbabwe and South Africa reported using traditional remedies and herbs to prevent COVID-19 infections.

COVID-19 does not exist. Narratives from participants who were unvaccinated revealed skepticism about COVID-19 as a real disease with fatal consequences. They stated:

“I have never heard that a Nyaope [type of drug] addict is sick with COVID.” 
*(UV_18–26 _GP)*


“I am also not concerned because the COVID symptoms are just in your head, and if you tell yourself that you have COVID, that’s when you’ll see symptoms; otherwise, you have to treat yourself.” 
*(UV_18–26_GP)*


Traditional herbs can cure COVID-19. While they did not see the need for the vaccine, they relied on traditional herbs as an alternative. One participant in the 18–24 age group shared how she tested positive for COVID-19 and was “cured” by isolating and taking herbs:

“[The] same thing that *Kim [Pseudonym used] said that she tested positive [for COVID-19], and I also tested positive before her and we took our herbs and quarantined, and we didn’t get sick. They called me after [a] month that I tested positive, and I did not vaccinate, but I was drinking the herbs, so I will not vaccinate.” 
*(UV_18–26_GP)*


A participant In the older group in the Western Cape stated that they would not vaccinate because they felt healthy, presumptively because they used traditional herbs:

“No, you will use your tablets or homemade medication like garlic and olive leaves. I still have garlic in my refrigerator. I don’t want to be vaccinated; I am healthy.” 
*(UV_50+_WC)*


A Zimbabwean participant said that, despite being vaccinated, they would take zumbani (an herbal tea and immune booster for the treatment or prophylaxis of various ailments, including influenza):

“Here I support what they are saying, that I can be vaccinated but I am safe if I get zumbani or whatever and just drink.” 
*(UV_45+_ZIM)*


### 3.3. Motivators for COVID-19 Vaccine Uptake

Motivators included accessibility of vaccination sites, trust in the government, experience with childhood vaccinations, and the belief that vaccines reduce the severity of COVID-19 symptoms. These were presented under the SAGE concept of confidence as a motivator. Fear of dying from COVID-19 and knowing someone who had died or was infected with COVID-19 were presented under complacency as a motivator. Lastly, accessibility of vaccination sites and user-friendly registration processes were presented under convenience as a motivator. 

#### 3.3.1. Convenience as a Motivator

Accessibility of COVID-19 vaccination sites. Most participants across the two countries were motivated to get vaccinated because the vaccination sites were located within their communities and mostly within walking distance or by using affordable transport: 

“I stay close to the clinic, and they would ask about where you stay and if you are able to come, and if not then they would tell you that on this particular day, they will be vaccinating at this particular place, which will be closer to where you stay, meaning you can get it easily.” 
*(V_18–44_ZIM)*


“Yes, because we have many vaccination sites that side, so it was not difficult at all.” 
*(V_18–35_GP)*


User-friendly registration process. In South Africa’s Gauteng province, a considerable number of participants in the 18–35 and 36–49 age groups reported the registration process to be “easy and friendly”. The online registration process provided information about the nearest vaccination site, which participants appreciated:

“It is easy, because you go there—even if you are not registered online, you go there, and they register you at the table. It is an easy process.” 
*(V_36–49_GP)*


#### 3.3.2. Confidence as a Motivator

Most vaccinated participants from both countries reported strong trust in the government and confidence in the COVID-19 vaccine. 

Trust in the government. A motivator for vaccinating was participants’ trust in their respective governments’ intentions for the vaccines. Some vaccinated and unvaccinated participants in Zimbabwe and Gauteng shared the perception that their government would not harm them using the vaccines. Two participants in the 18–44 age group and vaccinated—one from Zimbabwe and the other from Gauteng—supported the view that vaccines used by the government would not kill people. One stated:

“I am supporting [participant] number 46 and I personally felt that there’s no Ministry which brings a vaccine to kill us but for our survival because the nation will perish, so I did not.” 
*(V_18-44_ZIM)*


Participants rationalized that if previous vaccinations had been safe, then the COVID-19 vaccine would not pose any danger. Ultimately, these participants affirmed their trust in government’s intentions by getting immunized:

“The reason I got vaccinated is that I trust the government will not kill the people. I am still alive… the government and the department of health will not kill [the] people. If they say there is a disease, and they say we should vaccinate, surely they want us to live.” 
*(V_50+_GP)*


Confidence in the COVID-19 vaccine and its benefits. Participants’ perceptions included a belief in the vaccine’s ability to reduce the severity of COVID-19, and to protect against other disease. Participants perceived the COVID-19 vaccine as benefitting themselves and those who had been vaccinated. Although most responses in this theme came from vaccinated participants, a few were unvaccinated. 

A common sentiment among vaccinated participants in both countries was that vaccination reduced the severity of COVID-19 if infected. One participant from South Africa in the 50+ age group recounted how this viewpoint motivated them to vaccinate. Similarly, some vaccinated participants from Zimbabwe—one in the 50+ and two in the 18–44 age group—expressed the belief that the COVID-19 vaccine would protect them from severe illness should they become infected. They drew distinctions between illness prior to vaccination and after vaccination, affirming that the latter would be preferable:

“What really made me to get vaccinated is I heard that if you are vaccinated, it prevents the severity of COVID disease, and you experience less pain when it affects you because our immunity… they attack each other. Even my sister got COVID, and she had minor symptoms because she was vaccinated. Because of this, I decided to get vaccinated.” 
*(V_45+_ZIM)*


Some Zimbabwean participants reported that the COVID-19 vaccine protected against other diseases, such as a persistent cough, flu, and headache. Vaccinated participants in the 18–44 age range verified the benefits, having witnessed reduced severity of flu illnesses in their loved ones who had received the vaccine and become infected with COVID-19.

Experience with childhood vaccinations. Some participants in the study were motivated by their childhood vaccination experience. Some vaccinated participants from Zimbabwe, aged 45 years and older, also mentioned that they were motivated by “growing up vaccinated”. Their previous experiences with vaccinations shaped their positive views of vaccines. One stated:

“The other reason is we grew up having these vaccines, and we just know that, whenever something like this is introduced, it is beneficial in life.” 
*(V_45+_ZIM)*


#### 3.3.3. Low Complacency as a Motivator

In contrast with their unvaccinated counterparts, those who were vaccinated showed low complacency across all age groups and in both countries. The main reasons cited were mandatory vaccinations, and fear of dying from COVID-19 or knowing someone who had died or been infected with COVID-19. While some unvaccinated participants believed that COVID-19 did not exist, most reported having been exposed to someone who had COVID-19 or having been infected themselves. 

Fear of dying from COVID-19 infection. Low complacency was reported in the vaccinated group of participants who mentioned fear of dying from COVID-19 infection as one of the main motivators for getting vaccinated. Vaccinated participants in this study reported a high self-perceived risk of COVID-19 and the need to get vaccinated: 

“I vaccinated because I survived two times from Corona. I had Corona, so I saw a need to vaccinate since they said there will be [a] fourth wave, maybe I will not survive this time. So, I decided to get vaccinated to help myself because I nearly died as I had Corona two times. Yeah, I only wanted to stay alive.” 
*(V_36–49_GP)*


“What forced me to be vaccinated was I was afraid of dying and being crippled. I was scared, [and] that’s why I was vaccinated.” 
*(V_18-44_ZIM)*


Knowing someone infected with/or who had died from COVID-19. Some participants had witnessed someone close to them or in the community who was infected with or said to have died from COVID-19:

“I saw a person who was ill with COVID. He had difficulty in breathing, and you could feel pity for him. So, it made me realize that getting vaccinated is the best, and that’s what pushed me to get vaccinated.” 
*(V_18–44_ZIM)*


“I vaccinated because I survived two times from Corona. I had Corona, so I saw a need to vaccinate. Since they said there will be a fourth wave, maybe I will not survive this time. So, I decided to get vaccinated to help myself because I nearly died as I had Corona two times, I only wanted to stay alive.”- 
*(Participant 15, VIN_FGD_V_36-49 years_11-MAR-2022 GP)*


## 4. Discussion

Confidence, convenience, and complacency are critical concepts in understanding vaccine uptake in low- and middle-income countries. Overall, vaccine uptake in South Africa and Zimbabwe was influenced by inconvenience, a lack of confidence, and high levels of complacency toward the COVID-19 vaccine. Participants mistrusted the vaccine’s safety, and the perceived intentions of the governments of both countries, manufacturers, funders, and promoters. Trust is a second-order emotion, which assists people to “generate a constellation of emotions”, such as fear [25]. Lack of trust has been a common theme in several studies ranging from not knowing enough about the short-term and long-term dangers of the vaccine [26], to distrust of national governments’ ability to implement vaccination schemes or programs effectively [27].

Moreover, unvaccinated participants believed vaccines would do more harm than benefit their health. Confidence can be increased by providing transparent information about side-effects and adverse events, and establishing communication strategies that will ensure appropriately timed and culturally relevant dissemination of information. Improving convenience through effortless access to vaccination sites and vaccines, and by reducing queues, could increase vaccine uptake in South Africa and Zimbabwe. Most of the barriers highlighted from this study were relayed by unvaccinated participants, which may have contributed to their COVID-19 vaccine hesitancy. While many unvaccinated participants expressed negative attitudes, the findings also reveal positive perceptions toward COVID-19 vaccines, which may have motivated vaccine acceptance among vaccinated participants.

### 4.1. Confidence as a Barrier and Motivator

This study’s findings confirm vaccine hesitancy in the unvaccinated groups and greater vaccine confidence and uptake in the vaccinated groups in Zimbabwe and South Africa. These findings reinforce those of other studies in low- and middle-income countries [28,29]. Barriers to COVID-19 vaccination uptake included concerns around vaccine safety and efficacy. Participants were fearful of the vaccine’s side-effects, and of unknown adverse effects, including death, infertility, birth deficits, and reduced sexual pleasure. One study in Saudi Arabia found similar barriers to vaccine uptake around the potential adverse effects of certain vaccine technologies [30]. Such fears often stem from misinformation in the media or local communities, resulting in dwindling confidence in the COVID-19 vaccine. 

The main and most frequent barriers to COVID-19 vaccine hesitancy in a system review of studies by Cascini et al. (2021) were identified as the fear of safety and side-effects, the vaccine’s effectiveness and the rapid pace of its development when compared to others [31], such as polio, HPV. and flu vaccines, not to mention the HIV vaccine, which is still under process. The unprecedented speed of the COVID-19 vaccine’s development calls into question its safety and efficacy [13]. These factors were also identified as some of the main barriers to vaccine uptake in this study, in which participants’ misgivings about the apparent haste in developing the vaccine gave rise to feelings of fear and mistrust in its safety. 

Research shows that low vaccine uptake may be attributed to a lack of confidence in those recommending the vaccine. In a study by Gerretsen et al. (2021), participants mentioned mistrust in the COVID-19 vaccine’s development and suspicion of commercial profiteering by the government and vaccine manufactures [14]. The misinformation and conspiracy theories around COVID-19 vaccines so rife among unvaccinated individuals in South Africa and Zimbabwe are not altogether surprising given people’s mistrust and disillusionment with these countries’ governments, and perceptions that the government is coercing them to take the COVID-19 vaccine. 

Several studies have suggested that previous vaccination history may be a facilitator for COVID-19 vaccine acceptance and uptake. For instance, studies by Crawshaw et al. (2022) and Sherman et al. (2021) showed that individuals who have received the influenza vaccine were more likely to be vaccinated against COVID-19 [32,33]. The findings of the present study concur with these two studies—the concept of vaccinations for vaccine-preventable diseases is not new to low- and middle-income countries such as South Africa and Zimbabwe. The present study also found that familiarity with childhood vaccines put participants’ minds at ease when vaccinating for COVID-19. Some participants from Zimbabwe affirmed that getting vaccinated was self-evident given their past experiences with, for example, childhood polio and measles vaccines. This is consistent with models such as the integral behavior model, which postulates that knowledge, skills, and previous experience in performing the behavior have an important role. Furthermore, the 3C model of vaccine hesitancy hypothesizes that previous vaccination experience may bolster confidence, thereby encouraging vaccine uptake in some individuals [14]. 

### 4.2. Convenience as a Barrier and Motivator

In addition to the barriers mentioned in the discussions with unvaccinated participants, there are also motivators for vaccine uptake among vaccinated participants. Convenience as a motivator is crucial to vaccine uptake [14]. Studies postulate that accessible and convenient vaccination locations are a cost-effective intervention for increasing vaccine uptake, since they increase coverage and address equity concerns. These studies also found that pharmacy partnerships, mobile vaccination initiatives and pop-up clinics can reach populations in underserved areas [31,34]. Our findings reveal that effortless access to COVID-19 vaccines and local vaccination sites and straightforward registration processes contributed to vaccine uptake among participants. The present study found that, when participants lived within walking distance from clinics or did not have to spend much on transport, they would be more inclined to vaccinate. 

As seen in behavioral responses to other infectious pandemics such as Ebola, severe acute respiratory syndrome (SARS), and influenza, trust is an important determinant of the public’s acceptance of a government’s health crisis strategy, including vaccination [35]. Participants in the present study cited their faith in the government as a major reason for vaccinating and trusted that the government would not offer unsafe vaccines to its people. 

### 4.3. Complacency as a Barrier and Motivator

In this study, strong beliefs in traditional medicines as alternatives to vaccines diminished the acceptability of COVID-19 vaccines. Despite most unvaccinated participants believing that COVID-19 did not exist, some unvaccinated participants in Zimbabwe and South Africa believed in using traditional remedies and herbs to prevent COVID-19 infections. Steaming traditional roots and herbs (artemisia, garlic, and ginger) was the norm for some unvaccinated participants, which concurs with other studies [29]. In informal settlements in Lusaka, Pugliese et al. (2022) found that most people believed more in using traditional medicines than vaccines for vaccine-preventable childhood illness [29]. While participants in the present study did not specifically mention consulting with African traditional healers to cure or prevent COVID-19 symptoms and infections, the possibility is there. Therefore, integrating African traditional healers—deemed within communities to be an important part of the health care system—may be worth considering [36,37,38,39]. 

The desire to reduce the risk of severe symptoms and complications related to COVID-19 through vaccination was prominent among vaccinated participants aged 50+ and 18–44 in the Western Cape and Zimbabwe, respectively. Participants believed that should they contract COVID-19, their symptoms would be mild, and they would not die. A study on barriers to and facilitators of COVID-19 vaccines among incarcerated people in Canada reported similar results. Participants listed seeking both individual and collective protection against severe COVID-19 as a facilitator of vaccine acceptability. These participants self-identified as high risk owing to comorbidities associated with increased COVID-related morbidity and mortality, and they were of the view that the vaccine would protect against severe COVID-19 [40]. 

## 5. Limitations

Our study used qualitative methods, which implies that the data were specific to this specific study’s participants and cannot be generalized to other community members. Some participants may have been subject to response bias or influenced by the researcher or other participants in the FGD. Since we conducted an interview at only one timepoint, it is possible that participants’ views about vaccination have changed since then.

## 6. Conclusions

Vaccine hesitancy has massive potential to reduce vaccine uptake and diminish vaccine coverage [41] and is jeopardizing efforts to reach herd immunity. Therefore, understanding the underlying barriers to and motivators for vaccine uptake contributes toward more targeted health promotion communication strategies, which are so urgently needed to tackle vaccine hesitancy and support ongoing COVID-19 vaccine rollout programs. Local community factors impacting vaccine uptake confirm the pressing need for in-depth engagement of high-risk communities. Rather than making vaccinations mandatory, this points to the importance of augmenting efforts to educate the population on the benefits of vaccines and the risk associated with failure to vaccinate against vaccine-preventable diseases. 

This study has shown that COVID-19 vaccine uptake can be hampered by serious challenges, at both personal and structural levels. As such, there is a need to understand such barriers to bring about critical and necessary herd immunity. At the same time, it is possible to leverage on the motivators to enhance the uptake of vaccines. 

## Figures and Tables

**Figure 1 vaccines-11-00729-f001:**
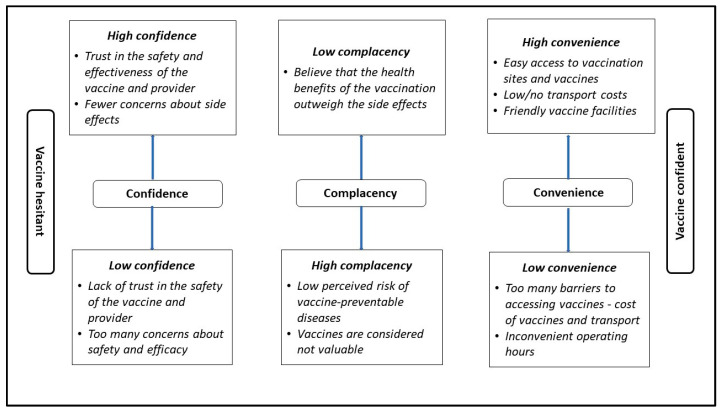
The 3Cs of vaccine hesitancy [10].

**Figure 2 vaccines-11-00729-f002:**
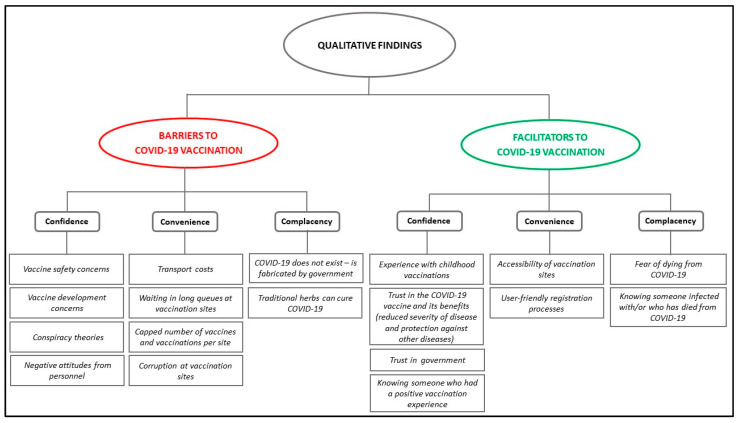
Findings using the SAGE 3C vaccine hesitancy model (adapted from the WHO Report of the SAGE Working Group on Vaccine Hesitancy [10]).

**Table 1 vaccines-11-00729-t001:** Biographic information of FGD participants in South Africa and Zimbabwe.

Age	South Africa	Zimbabwe	Total
(Years)	n (%)	n (%)	n (%)
18–19	–	6 (12.5)	6 (4.7)
20–29	22 (27.2)	8 (16.7)	30 (23.2)
30–39	18 (22.2)	10 (20.8)	28 (21.7)
40–49	13 (16.0)	17 (35.4)	30 (23.2)
50–59	22 (27.2)	3 (6.3)	25 (19.4)
60+	6 (7.4)	4 (8.3)	10 (7.8)

## Data Availability

Qualitative transcripts may contain identifying information. Therefore, data will be made available upon request to the corresponding author.

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
