# Peer review of "A Qualitative Study Exploring Motivators and Barriers to COVID-19 Vaccine Uptake among Adults in South Africa and Zimbabwe"

_vaccines, 2023, doi:10.3390/vaccines11040729_

Round 1
Reviewer 1 Report
I have revised the paper entitled "A qualitative study exploring motivators and barriers to 2 COVID-19 vaccine uptake among adults in South Africa and 3 Zimbabwe" . I think it is interesting for the audience of this journal. I reccomend to give more focus on the concept of the trust and confidence following the examples contained in these papers:
- Bates, B. R., Villegas-Botero, A., Costales, J. A., Moncayo, A. L., Tami, A., Carvajal, A., & Grijalva, M. J. (2022). COVID-19 vaccine hesitancy in three Latin American countries: Reasons given for not becoming vaccinated in Colombia, Ecuador, and Venezuela. Health Communication, 37:12, 1465-1475. https://doi.org/10.1080/10410236.2022.2035943
- Belli, S., & Broncano, F. (2017). Narratives of trust: sharing knowledge as a second-order emotion. Human Affairs, 27(3), 241-251.
- Gewirtz-Meydan, A., Mitchell, K., Shlomo, Y., Heller, O., & Grinstein-Weiss, M. (2022). COVID-19 among youth in Israel: Correlates of decisions to vaccinate and reasons for refusal. The Journal of Adolescent Health: Official Publication of the Society for Adolescent Medicine, 70(3), 396–402. https://doi.org/10.1016/j.jadohealth.2021.11.016
Author Response
have revised the paper entitled "A qualitative study exploring motivators and barriers to 2 COVID-19 vaccine uptake among adults in South Africa and 3 Zimbabwe" . I think it is interesting for the audience of this journal. I recommend to give more focus on the concept of the trust and confidence following the examples contained in these papers:
- Bates, B. R., Villegas-Botero, A., Costales, J. A., Moncayo, A. L., Tami, A., Carvajal, A., & Grijalva, M. J. (2022). COVID-19 vaccine hesitancy in three Latin American countries: Reasons given for not becoming vaccinated in Colombia, Ecuador, and Venezuela. Health Communication, 37:12, 1465-1475. https://doi.org/10.1080/10410236.2022.2035943
- Belli, S., & Broncano, F. (2017). Narratives of trust: sharing knowledge as a second-order emotion. Human Affairs, 27(3), 241-251.
- Gewirtz-Meydan, A., Mitchell, K., Shlomo, Y., Heller, O., & Grinstein-Weiss, M. (2022). COVID-19 among youth in Israel: Correlates of decisions to vaccinate and reasons for refusal. The Journal of Adolescent Health: Official Publication of the Society for Adolescent Medicine, 70(3), 396–402. https://doi.org/10.1016/j.jadohealth.2021.11.016
Response: Thank you for this comment and the suggested references. We have incorporated the references and expanded on the themes of trust and confidence in the introduction and discussions sections as below on page 11 on lines 488-493
Please see below the added text:
Trust is considered a second-order emotion, which assists people to “generate a constellation of emotions”, such as fear [23]. Lack of trust has been a common theme in several studies ranging from not knowing enough about short-term and long-term harms of the vaccine [24], to distrust of national governments’ ability, to effectively implement vaccination schemes or programs [25].
Reviewer 2 Report
In this article, the authors analize hesitancy to COVID-19 vaccine in South Africa and Zimbabwe.
The title of the article is apposit, the keywords are well chosen, the abstract is clear and appropriate in lengh and content.
The introdution is sufficient to provide the necessary background, but it references only 4 articles, so it may be a good idea to include some more to emphasize the interest for this type of studies.
The references are relevant and recent enough (please adfd the year to the "accessed on" statements).
Materials and methods' description is clear and detailed. I suggest to add an explanation of the meaning of the stratification by HIV status.
The tables' layout is clearly readable and their labelling is apposit. The reference to Table 1 in the first paragraph of 2.5 should be to Table 1 and Table 2 instead.
The discussion of data is lucid and the conclusions are consistent. I let to the editor to judge if the example sentences from the participants should be part of the main article opr moved to supporting info.
The limitations of the study are honestly and clearly reported.
The novelty of this study is not so high, but the geographical specificity may help to better understand the rational of vaccine hesitancy in a critical area, and therefore it may be of potential interest for the readers of the journal.
So in my opinion it is, all considered, suited for publication (after the suggested editing).
Author Response
In this article, the authors analize hesitancy to COVID-19 vaccine in South Africa and Zimbabwe.
The title of the article is apposit, the keywords are well chosen, the abstract is clear and appropriate in lengh and content.
The introduction is sufficient to provide the necessary background, but it references only 4 articles, so it may be a good idea to include some more to emphasize the interest for this type of studies.
Response: Thank you for these comments and the suggestions to improve the manuscript. Additional articles have been cited in the introduction.
The references are relevant and recent enough (please add the year to the "accessed on" statements).
Response: The year for references with an Access Date has now been included to the reference list.
Materials and methods' description is clear and detailed. I suggest to add an explanation of the meaning of the stratification by HIV status.
Response: An explanation was included around the stratification by HIV status and conducting FGDs. The following was included, see lines 145-151:
This stratification was based on the local context and what was deemed appropriate based on the local study team’s feedback. Therefore, in Zimbabwe there was a separate group for individuals living with HIV and in South Africa FGDs were not stratified by HIV status, due to the stigma related to HIV. Focus group discussions allowed for collection of contextual data that was used to gain an in-depth understanding of the motivators and barriers to COVID-19 vaccine uptake [19].
The tables' layout is clearly readable and their labelling is apposit. The reference to Table 1 in the first paragraph of 2.5 should be to Table 1 and Table 2 instead.
Response: Thank you for picking up this oversight. This has been corrected. Please see lines 153 –154: All participants completed a socio-demographics questionnaire assessing age (Table 1), gender, level of education, language spoken at home and employment status (Table 2).
The discussion of data is lucid and the conclusions are consistent. I let to the editor to judge if the example sentences from the participants should be part of the main article opr moved to supporting info.
Response: For qualitative manuscripts. the quotations provide supporting evidence and typically the quotations are in the main manuscript.
Reviewer 3 Report
1. Introduction section:
I suggest adding more information about types of vaccines available to be more interesting also I suggest to begin the introduction by summarizing the negative impact of COVID-19 and justify the need for vaccination.
2. Conclusion section:
Please link the results to the conclusion.
Author Response
Response: Thank you for these comments and suggestions.
Introduction Section
I suggest adding more information about types of vaccines available to be more interesting also I suggest to begin the introduction by summarizing the negative impact of COVID-19 and justify the need for vaccination.
Response: We have added a list of the vaccines that were recognized by the WHO. The following paragraph has been included to the introduction (Page 2,lines 58-64): COVID-19 vaccines recognized by the World Health Organisation (WHO) include Pfizer–BioNTech, Oxford–AstraZeneca, Sinopharm BIBP, Moderna, Janssen, CoronaVac, Covaxin, Novavax, Convidecia, and Sanofi–GSK [7]. Seven others are under assessment by the WHO: Sputnik V, Sinopharm WIBP, Abdala, Zifivax, Corbevax, COVIran Barekat, and SCB-2019 [8] . Vaccines are created to prevent the spread of an infectious disease and in the process achieve herd immunity in a population.
2. Conclusion section:
Please link the results to the conclusion.
Response: The conclusion has been revised to strengthen the link to the results. The following summary paragraph has been included, see page 14, lines 606-609: This study has shown that COVID-19 vaccine uptake can be hampered by serious challenges, both at personal and at structural levels as such there is need to understand such barriers in order to bring about critical and necessary herd immunity. At the same time, it is possible to leverage on the motivators to enhance uptake of vaccines.
Reviewer 4 Report
This article is based on a qualitative study that explores motivators and barriers to COVID-19 vaccine uptake among adults in South Africa and Zimbabwe. Authors used an exploratory qualitative research design by conducting 18 focus group discussions and conceptualized the key themes within the World Health Organization’s Strategic Advisory Group of Experts on Immunization (SAGE) 3Cs (convenience, confidence and high complacency) vaccine hesitancy model. They found that vaccine uptake was influenced by inconvenience, lack of confidence, and high levels of complacency towards COVID-19 vaccine. Therefore, in their opinion is possible counteract vaccine hesitancy improving confidence and convenience: providing transparent information about side effects; establishing communication strategies; improving effortless access to vaccination, etc.
Overall, the study is appealing and the manuscript is of potential interest for readers of Vaccines.
My comments are listed below:
-Why Authors didn’t consider the more recent and specific 5C model or the extended 7C model?
-In line 111 is reported “A total of 18 FDGs, each with a minimum of eight participants…” it should be at least 144 participants instead in Table 2 a total of 129 p
Author Response
This article is based on a qualitative study that explores motivators and barriers to COVID-19 vaccine uptake among adults in South Africa and Zimbabwe. Authors used an exploratory qualitative research design by conducting 18 focus group discussions and conceptualized the key themes within the World Health Organization’s Strategic Advisory Group of Experts on Immunization (SAGE) 3Cs (convenience, confidence and high complacency) vaccine hesitancy model. They found that vaccine uptake was influenced by inconvenience, lack of confidence, and high levels of complacency towards COVID-19 vaccine. Therefore, in their opinion is possible counteract vaccine hesitancy improving confidence and convenience: providing transparent information about side effects; establishing communication strategies; improving effortless access to vaccination, etc.
Overall, the study is appealing and the manuscript is of potential interest for readers of Vaccines.
Response: Thanks for the review, comments and suggestions. We have the following responses:
-Why Authors didn’t consider the more recent and specific 5C model or the extended 7C model?
In this study, we used the 3 C’s model as opposed to the 5 C’s which includes confidence, complacency, constraints, calculations, and collective responsibility. Neither did we use the more recent 7 C’s model which includes confidence, complacency, constraints, calculation, collective responsibility, compliance, and conspiracy.
Response: The following has been included to the introduction, page 2, lines 70-74: In this study, we used the 3 Cs model as opposed to the 5 Cs, which includes confidence, complacency, constraints, calculations, and collective responsibility [10]. Neither did we use the more recent 7 Cs model which includes confidence, complacency, constraints, calculation, collective responsibility, compliance, and conspiracy [11] .
-In line 111 is reported “A total of 18 FDGs, each with a minimum of eight participants…” it should be at least 144 participants instead in Table 2 a total of 129 p.
Response: Thank you for picking this up, we had a total of 129 participants and the sentence in lines 131-132 was revised as follows: A total of 18 FGDs were conducted, with a minimum of seven participants each in 15 of the 18 FGDs and three FGDs having 8 participants each.